# The Association between Vision Impairment and Depression: A Systematic Review of Population-Based Studies

**DOI:** 10.3390/jcm11092412

**Published:** 2022-04-25

**Authors:** Gianni Virgili, Mariacristina Parravano, Davide Petri, Erica Maurutto, Francesca Menchini, Paolo Lanzetta, Monica Varano, Silvio Paolo Mariotti, Antonio Cherubini, Ersilia Lucenteforte

**Affiliations:** 1Department of Neurosciences, Psychology, Drug Research and Child Health (NEUROFARBA), University of Florence and AOU Careggi, 50139 Florence, Italy; gianni.virgili@unifi.it; 2IRCCS—Fondazione Bietti, 00198 Rome, Italy; mariacristina.parravano@fondazionebietti.it (M.P.); monica.varano@fondazionebietti.it (M.V.); 3Department of Clinical and Experimental Medicine, University of Pisa, 56126 Pisa, Italy; davide.Petri@unipi.it; 4Department of Medicine—Ophthalmology, University of Udine, 33100 Udine, Italy; ericamaurutto@libero.it (E.M.); francescamenchini@gmail.com (F.M.); paolo.lanzetta@uniud.it (P.L.); 5NCD Department, World Health Organization, 1211 Geneva, Switzerland; mariottis@who.int; 6Geriatria, Accettazione Geriatrica e Centro di Ricerca per L’invecchiamento IRCCS INRCA, 60124 Ancona, Italy; a.cherubini@inrca.it

**Keywords:** blindness, depression, meta-analysis, systematic review, visual impairment

## Abstract

We conducted a systematic review and meta-analysis to investigate whether depression is associated with vision impairment (VI) in population-based studies in adults. MEDLINE and EMBASE were searched, from inception to June 2020. Studies were included if they provided two-by-two data for calculating the OR of association between VI and depression, or crude and/or an adjusted odds ratio (OR) with a corresponding 95% confidence interval (CI) were reported. The proportion of VI and depression was also extracted. ORs were pooled using random-effect models, proportions were pooled using random intercepts logistic regression models. Overall, 29 articles (31 studies) were included: of those, 18 studies used survey data (622,312 participants), 10 used clinical examination data (69,178 participants), and 3 used administrative databases (48,162,290 participants). The proportion of depression (95%CI) was 0.17 (0.13–0.22) overall and 0.27 (0.21–0.33) in VI subjects. The proportion of VI was 0.10 (0.07–0.16) overall and 0.20 (0.13–0.29) in depressed subjects. The association between VI and depression was direct: crude ORs were 1.89 (1.51–2.37) for survey data, 2.17 (1.76–2.67) for clinical examination data, and 3.34 (1.01–11.11) for administrative databases; adjusted ORs were 1.75 (1.34–2.30), 1.59 (1.22–1.96), and 2.47 (0.97–6.33), respectively. In conclusion, VI and depression are prevalent morbidities and should be actively sought when either is identified, especially in older adults.

## 1. Introduction

Vision loss is associated with reduced performance in activities of daily living, with a greater risk of falls, social isolation, institutionalisation, and even death [1]. A recent systematic review showed that hearing and vision loss are associated with frailty among adults living in the community [2]. Depression is a major morbidity in older adults and has been shown to be even more prevalent in individuals with vision impairment (VI). Depression of any grade has been found in one-third of visually impaired older adults, approximately twice as high as the lifetime prevalence rates in the normal-sighted older population, where the prevalence of depressive symptoms is about 15% [3,4,5,6,7,8,9]. Depression in itself affects the quality of life, even when symptoms are only mild [7,10,11].

Depressed patients with poor vision may experience more disability, since depression can also have an impact on one’s own perceived motivation and determination, toward achieving goals [12]. Moreover, depression is treatable, at least partly, and numerous mental healthcare programs have been implemented and found beneficial in eye care settings [13]. Additionally, a recent Cochrane systematic review, aiming to assess the effect of different psychological interventions on quality of life in low-vision patients, pooled the results of several studies and showed that psychological therapies or group programs may improve depression and enhance self-esteem in low vision adults [14].

A PUBMED and PROSPERO search (((depression OR depressive) AND (vision OR visual))) revealed that, to date (30 March 2022), there have been no recent reviews or meta-analyses on the burden of the association between VI and depression in the general population. Our study aimed to investigate whether depression is associated with VI in community-dwelling adults. The results of our study could be important to develop public health strategies to improve the quality of life of older adults. 

## 2. Materials and Methods

We performed a systematic review and meta-analysis according to the recommendations indicated by the Cochrane network [15]. The reporting was in accordance with the Preferred Reporting Items for Systematic Reviews and Meta-Analyses (PRISMA) criteria [16].

### 2.1. Inclusion Criteria

We sought to include population-based cross-sectional studies of community-dwelling adults. Case–control studies and randomised controlled trials were excluded, as were studies conducted in nursing homes, or in eye clinics and low-vision services, which were objects of a previous study [17]. 

We accepted the diagnostic category of VI as applied by the authors of each study, which included the World Health Organisation (WHO) definition, US definitions of VI and blindness, self-reported low-vision, registry-based definitions, including administrative/insurance data. For studies using clinical examination, we had planned to use the International Classification of Diseases 11 [18] to create study subgroups according to VI severity, defined according to best-corrected visual acuity (BCVA) in the better eye as mild (<6/12), moderate to severe (<6/60), blindness, (<3/60). However, the number of these studies was small, and most used mild low-vision or worse as the primary definition of VI.

We accepted the definition of depression as reported and implemented by study authors, including validated questionnaires, diagnostic criteria of the *Diagnostic and Statistical Manual of Mental Disorders* (DSM III or IV) or psychiatric specialist assessment, and self-reported depression. 

Studies were included if they provided two-by-two data for calculating the ORs of association between VI and depression, as defined by the investigators; studies were also included if raw data were not available, but a crude and/or an adjusted odds ratio (OR) with corresponding 95% confidence interval (CI) were reported. When possible, the frequency of VI and depression was also extracted and pooled. The protocol for this review is available from the authors upon request.

### 2.2. Search Strategy

The electronic databases MEDLINE (via PUBMED) and EMBASE were searched from inception to 7 June 2020. The search strategies are shown in detail in Appendix A.

Four independent reviewers (E.M., F.M., D.P., M.P.) analysed the output of the search and selected the studies, with a duplicate classification of literature and data extraction to ensure accuracy. Discrepancies were solved by an agreement between reviewers or with a senior reviewer (G.V.). 

### 2.3. Risk of Bias Assessment

All included studies were subjected to methodological critical appraisal by means of an adapted risk of bias tool for prevalence studies [19]. A maximum of 10 stars was assigned in 5 domains, with a maximum of 2 stars for each domain. The domains analysed in this study were lack of generalisability bias, record bias, attrition bias, detection bias, and reporting bias. The working definitions to assess bias are presented in the Appendix A.

### 2.4. Statistical Methods

We presented data by type of data source since we believe this is a major source of heterogeneity and is also critical to applicability. 

We pooled study-specific ORs using the ‘metagen’ command in R software (R Foundation) as follows: random-effect models were performed using inverse variance methods for pooling, Der Simonian–Laird estimator for tau^2^, and the Jackson method for confidence intervals of tau^2^ and tau. Study-specific proportions were pooled with the ‘metaprop’ command in R software (R Foundation) as follows: we fitted random intercept logistic regression model, used maximum-likelihood estimator for tau^2^, logit transformation of proportions, and Clopper–Pearson CI for individual studies. Pre-planned heterogeneity investigation was based on age 65 years or more, the inclusion of people with systemic comorbidities, the type of questionnaires used for diagnosing depression, and the definition of VI. Heterogeneity was assessed graphically and by inspecting I-square and reporting the predictive interval for primary meta-analyses.

## 3. Results

### 3.1. Results of Searches

Our search found 7053 records; of those, 6896 articles were excluded because they did not investigate depression in subjects with VI (Figure 1). Of the 157 remaining articles, 101 were rejected for other reasons, including full-text not in English, insufficient data, presence of subthreshold depression, unclear sampling, identical study population to other included studies, and longitudinal design. Consequently, 56 remaining full-text articles investigated depression status in patients with low vision. We excluded 27 studies which were conducted in eye clinics or low-vision services and had been included in another review [17]. Thus, we included 29 studies which were population-based. One study [20] reported adjusted ORs of association between VI and depression separately by men and women and was considered as providing two datasets for analyses. Another study [21] included two datasets collected in different periods. Finally, 31 independent datasets (29 studies), hereby referred to as ‘studies’, were included in this review.

### 3.2. Characteristics of Included Studies

A summary of the main characteristics of the 31 studies [3,20,21,22,23,24,25,26,27,28,29,30,31,32,33,34,35,36,37,38,39,40,41,42,43,44,45,46,47] is presented in Table 1. The number of participants ranged from 218 to 48,583,771. Overall, 10 studies [3,22,23,25,28,30,35,38,43,45] were conducted in Europe, 10 [21,24,29,32,34,37,39,42,47] in the United States, 9 [20,26,27,33,40,41,44,46] in Asia, 1 study [31] included data both from 2 European and 2 Northern American sites, and 1 study [36] was conducted on Cuban American, Mexican American, and Puerto Rican adults. Overall, 18 studies [20,21,24,26,29,31,33,35,36,37,39,41,42,43,44,46] were based on survey data (622,312 participants); 10 studies [3,22,23,25,27,30,34,38,45,47] used clinical examination (69,178 participants); 3 studies [28,32,40] used administrative databases (48,162,290 participants).

In total, 22 studies [3,21,22,23,25,26,28,29,31,33,34,35,36,38,40,41,42,43,45,46,47] presented raw data with a cross-tabulation of a dichotomous definition of VI and depression. Two studies [24,39] did not provide such data and reported only crude OR of this association, three studies [20,37] reported only adjusted OR, using age/gender and socioeconomic status and/or comorbidities, and four studies [27,30,32,44] reported crude and adjusted OR.

The included studies used a range of VI definitions, and in 15 studies [20,21,22,23,24,29,30,37,39,41,42,43,46], VI was self-reported (Table 1 and Appendix A). Twelve studies [3,25,26,27,33,34,35,36,38,44,45,47] used clinical examination defined VI based on better-eye BCVA cut-offs between 20/40 and 20/63. Two studies [32,40] based on electronic data used VI diagnosis based on the International Diseases Code (ICD-9 or 10), one study [28] used Read Codes, which are the morbidity coding system used in all UK primary care medical records, and one study [31] used clinical records.

The tools adopted to detect depression also varied: the Centre for Epidemiologic Studies Depression (CES-D) Scale was applied in seven studies [20,21,34,36,42,45]; the Geriatric Depression Scale-15 (GDS-15) was used in six studies [3,26,33,41,44,46]; the Patient Health Questionnaire-9 (PHQ-9) was used in four studies [21,37,43,47]. Other depression diagnoses were based on the International Classification of Disease (ICD-9 or ICD-10) [22,32,40], Beck’s Depression Inventory [25], the Zung Depression Status Inventory [38], the Hamilton Rating Scale [39], the Word Health Organisation Composite International Diagnostic Interview [30], the Mental Health Inventory Screening Test-5 [23], and the Depression Rating Scale [31]. In one study, depression was established by a Read code [28], while self-reported depression was accepted in four studies [24,27,29,35].

Appendix A summarises the risk of bias evaluation of all studies, comprehensive of lack of generalisability bias, record bias, attrition bias, detection bias, and reporting bias. Most studies had a score of 1 or 2* in the major domains of the quality scale used. Overall, 14 studies [3,21,25,26,32,33,34,36,37,39,41,42,47] reached a total score between 8 and 10, while 17 reached a total score between 6 and 7. Generalisability bias reached a score of 2 in 11 studies [21,25,26,27,28,32,33,36,41,47] and 1* in 20. Lack of clarity concerning the definition of VI and/or depression was the main quality issue in this domain.

About half of the studies (19 out of 31) [3,21,22,23,25,26,27,30,33,34,35,38,39,41,42,44,46,47] obtained a score of 2* for record bias, whereas the other studies collected data retrospectively or from registries. In total, 18 studies [3,21,24,25,26,28,29,31,32,34,36,37,39,40,41,42,43] were also rated with 2* regarding attrition bias, and 13 studies gained 1* because more than the 10% of the eligible subjects were not included. Only seven studies [21,26,35,36,37,41] reached 2* for detection bias, while no information was available in all other studies regarding masking of depression assessment with respect to vision status and were assigned 1*. Regarding reporting bias, we assigned 1* to 9 studies, and 22 studies [3,20,22,23,25,30,31,32,33,34,37,38,39,40,41,42,43,44,45,46,47] gained a score of 2*. There was one study [28] with zero stars due to the lack of details in the diagnostic process of depression declared.

### 3.3. Findings

Table 2 presents the meta-analysis of proportions of depressed participants, overall and among VI subjects, as well as the proportion of VI participants, overall and among depressed subjects. Appendix A present the corresponding forest plots.

The pooled *proportion of depression* was 0.17 (95% confidence interval (CI): 0.13–0.22) with predictive interval 0.03 to 0.56 (Table 2 and Appendix A). This proportion did not differ (*p*-value = 0.0664) by data type: 0.19 (0.14–0.27) for survey data (17 studies, 583,563 participants), 0.15 (0.10–0.23) for clinical examination data (8 studies, 36,167 participants), and 0.07 for administrative databases (2 studies, 309,948 participants). As is common with meta-analyses of prevalence data, heterogeneity was very high (I-square > 99%) overall and within subgroups.

The pooled *proportion of VI* was 0.10 (0.07–0.16), with predictive interval 0.01 to 0.64 (Table 2 and Appendix A). This proportion differed by data type (*p*-value = 0.0018): 0.16 (0.10–0.25) for survey data (16 studies, 611,430 participants), 0.07 (0.03–0.13) for clinical examination data (9 studies, 64,559 participants), and 0.03 (0.01–0.07) for administrative databases (3 studies, 47,892,290 participants). Heterogeneity was very high (I-square > 99%) overall and within subgroups.

The estimate of *proportion of depression in VI subjects* was 0.27 (0.21–0.33) with predictive interval 0.07 to 0.64 (Table 2 and Appendix A). This proportion did not differ by data type (*p* = 0.7868): 0.29 (0.22–0.37) for survey data (10 studies, 82,804 VI participants), 0.26 (0.17–0.38) for clinical examination data (8 studies, 3323 VI participants), and 0.22 (0.08–0.46) for administrative databases (3 studies, 480,029 VI participants). Heterogeneity was very high (I-square > 96%) overall and within subgroups.

The estimate of *proportion of VI in depressed subjects* was 0.20 (0.13–0.29) with predictive interval 0.02 to 0.73 (Table 2 and Appendix A). This proportion did not differ by data type (*p*-value = 0.1387): 0.28 (0.16–0.43) for survey data (10 studies, 89,897 depressed participants), 0.15 (0.09–0.25) for clinical examination data (8 studies, 3459 depressed participants), and 0.08 (0.02–0.33) for administrative databases (2 studies, 36,692 depressed participants). Heterogeneity was very high (I-square > 96%) overall and within subgroups.

Table 3 presents the association between VI and depression. The unadjusted association between VI and depression was OR (Appendix A): 1.89 (1.51–2.37) for survey data (14 studies, 617,355 participants), 2.17 (1.76–2.67) for clinical examination data (10 studies, 69,178 participants), and 3.34 (1.01–11.11) for administrative databases (3 studies, 48,162,290 participants). Heterogeneity was moderate for clinical examination (I-square: 66%) and high for survey and administrative data (I-square > 89%). The adjusted association between VI and depression was (Appendix A): OR 1.75 (1.34–2.30) for survey data (eight studies, 50,437 participants), 1.17 (1.00–1.37) for clinical examination data (four studies, 57,352 participants), and 2.47 (0.97–6.33) for administrative databases (two studies, 48,143,511 participants). Heterogeneity was nil for clinical examination (I-square: 0%) and high for survey data and administrative databases (I-square > 99%).

### 3.4. Subgroup Analysis

We evaluated whether study-level variables modified the association of VI with depression, using crude ORs. Differences between diagnostic depression instruments were difficult to assess statistically because of the large number of tools (Appendix A). Overall, pooled estimates varied across subgroups (*p*-value = 0.0170), with the estimate for CESD, PHQ-p, and GDS being superior to the pooled OR values for self-reported and other tools. There was no difference in the strength of the association of VI with depression between 17 studies with a lower age limit of 50–65 years (2.18, 1.72–2.75, Appendix A), compared with 10 studies including all adults (2.10, 1.33–3.32).

## 4. Discussion

Our systematic review confirms that VI and depression are common coexistent morbidities in adults and older people in the general population. This association was maintained in most studies regardless of the prevalence of either condition, which varied widely, as well as of the methods used to detect these conditions. In general, surveys found a higher prevalence of both VI and depression, compared with clinical examination. Studies based on electronic records yielded variable results but still a consistent association between depression and VI.

Of interest is subgroup analyses that suggested a similar, strong association in studies including younger adults. Although we found overall significant differences between diagnostic depression tools, we were unable to draw conclusions on the potential difference in their diagnostic sensitivity, due to the small number of studies in each group and methodological differences.

Our strategy for study selection was broad and about 2500 titles and 150 full-text articles were screened for eligibility. Despite high heterogeneity, which is expected, at least for prevalence data, due to variability in the definitions used, pooled estimates were consistent. However, the large majority of the included studies were based in high-income countries; thus, our results may not generalise to other contexts. A recent study investigated this issue in five low-middle income settings, including two national surveys (Guatemala, Maldives) and three regional/district surveys (Nepal, India, Cameroon). Although dichotomous data for the association between depression and VI could not be extracted, the study demonstrated increased adjusted odds of severe depression and severe anxiety among adults with mobility, hearing, and visual functional difficulties in all settings, with ORs ranging from 2.0 to 14.2 [48].

The prevalence of depression in community-dwelling individuals varies across studies, with systematic review pooled estimates between 13.3% [49] and 28.4% [50] depending on the inclusion of major depression only, rather than also of preclinical depression, respectively. In 2019, the worldwide prevalence of VI and depressive disorders were 9.6% and 3.8%, respectively [51]. Furthermore, a bidirectional association between VI and depressive disorders has been reported [51]. Specifically, a large population-representative survey of 7548 participants [52] found that baseline self-reported vision loss was significantly associated with future reports of depression (hazard ratio (HR), 1.33; 95%CI, 1.15–1.55) and also that, conversely, baseline depression was significantly associated with future reports of self-reported VI (HR, 1.37; 95%CI, 1.08–1.75). This pattern was confirmed for both VI and hearing loss by Liu et al. [53], who concluded that the mental health of people with these disabilities should be the focus, and regular assessments of vision and hearing in people with depressive symptoms are recommended.

Our results are in agreement with those reported in a previous systematic review published by our group including 27 studies on patients attending eyes clinics and low vision rehabilitation services [17]. The prevalence of depression was slightly higher in patients attending clinics than in community-dwelling adult patients (0.25 vs. 0.17, respectively). Another systematic review published in 2015 and aimed at investigating the relationship between VI and depression in the elderly showed an association in all 10 included studies [54]. The review, however, did not report a pooled proportion and concluded that ‘we could not definitely establish an association between VI and depression in the elderly. The reasons include a lack of standardised measures of VI and depression to allow comparability of the studies and the potential bias created by several other variables’.

The implications of our results with regard to people-centred health care concern multidisciplinary and integrated pathways of care, linking general practitioners with ophthalmologists and psychiatric/psychological professionals, which should be considered, given the frequency of this type of comorbidity [13,14,55,56,57,58]. A recent review has linked sensory loss, including at least hearing and visual loss, with frailty [2]. The attenuation of the association between VI and depression found by us with adjusted vs. crude ORs is strongly suggestive of shared effects with other systemic conditions, either through common risk factors or with a reciprocal enhancement of these disabilities [2].

Our review of observational studies aimed to investigate the burden of multimorbidity regarding two prevalent conditions but cannot prove a causal relationship between VI and depression, as stated above. Despite this limitation, case-finding in clinical [17] or population-based studies can be useful since several types of psychological interventions, mainly behavioural cognitive therapy, as well as methods for enhancing vision were found to slightly improve depression in adults with VI in a recent Cochrane review of randomised controlled studies [17].

We also acknowledge as a limitation the fact that the definitions of VI and the cut-offs used for depression diagnosis differed among studies. Regarding the risk of bias, adequate recruitment of consecutive patients was recorded in only about half the studies. In addition, whether studies used a masked assessment of depression and VI was unclear in about half the studies. Finally, about half the studies were at risk of record bias since they were retrospective or registry-based.

A potential limitation of our review is that, despite our effort to be comprehensive by taking a broad approach, we may have missed studies, given that depression data may be included as a secondary outcome in studies with different overall objectives. Another limitation may be that we did not include longitudinal studies. However, we made this choice since we did not aim to explore causality but rather to investigate the burden of co-presence of VI and depression in the general population. Longitudinal studies have proven that VI is associated with incident depression and highlighted that this can be due to difficulties with reading, mobility, and driving [59,60,61]. However, studies also found that baseline depression predicts the development of VI [52]. The complex relationship between VI and depression should be considered in the broader context of multiple associations between ocular diseases and systemic conditions, specifically neurological diseases. The increasing understanding of shared causal factors is also fundamental to clarifying such complexity. Previous studies have demonstrated that eye diseases, particularly AMD, may be associated with Alzheimer’s and Parkinson’s disease [62,63]. On the other hand, smoking, alcohol intake, and reduced physical exercise are also risk factors for AMD, Alzheimer’s, and Parkinson’s disease [64,65,66,67,68], and also for cardiovascular disease and cancer. Understanding shared causality among complex chronic conditions may need further research. Our study highlights that VI can be used as a marker of multimorbidity and frailty in older patients, as well as the need for an integrated approach in the delivery of people-centred care, as preconised in the WHO’s ICOPE framework. This association was consistent regardless of the method used to detect VI, i.e., clinical examination, interview, questionnaire, or routinely collected data.

## 5. Conclusions

In conclusion, one in five subjects with depression has VI on clinical assessment and one in four with survey data. These prevalent morbidities should be actively sought when either is identified, especially in older adults.

## Figures and Tables

**Figure 1 jcm-11-02412-f001:**
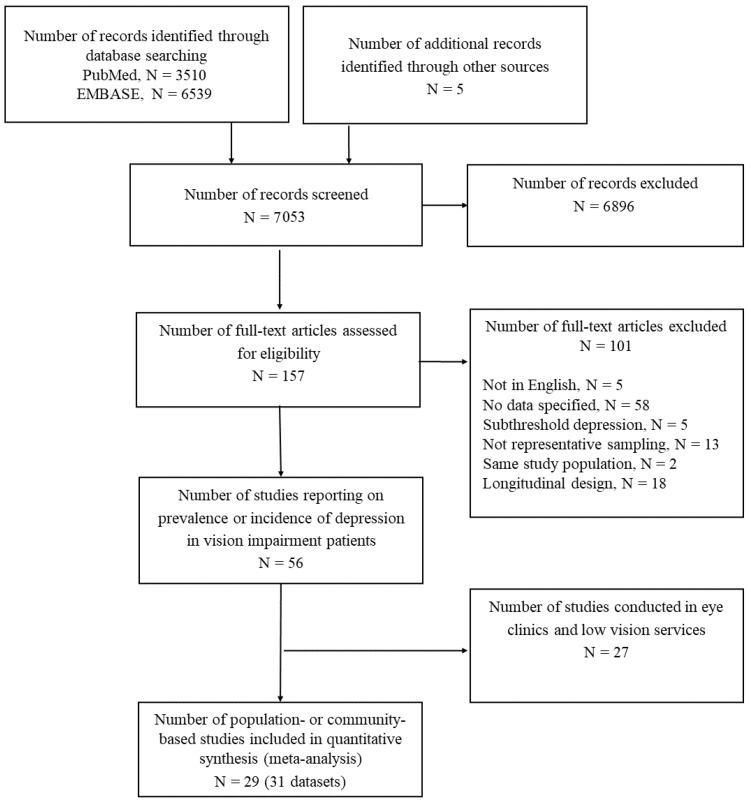
Flow diagram summarising the process for selecting original articles for review.

**Table 1 jcm-11-02412-t001:** Characteristics of included studies.

	N° of Studies	N° of Subjects (Range)
Overall	31	48,853,780 (218–47,852,342)
Country		
United States	10	47,917,779 (484–47,852,342)
Europe	10	328,989 (218–291,169)
Asia	9	54,220 (254–28,392)
Mixed	2	552,792 (2432–550,360)
Data source		
Survey	18	622,312 (218–550,360)
Administrative database	3	48,162,290 (18,779–47,582,342)
Clinical examination	10	69,178 (437–28,392)
Criteria for visual impairment		
Self-reported	15	80,100 (400–36,110)
BCVA	12	61,030 (218–28,392)
ICD	2	47,871,121(18,779–47,852,342)
Other criteria	2	841,529 (291,169–550,360)
Criteria for depression		
CES-D Scale	7	11,447 (484–2,591)
GDS-15	6	17,118 (254–13,900)
PHQ-9	4	22,782 (567–10,480)
Self-reported	4	74,552 (218–36,110)
Other criteria	10	48,727,881 (437–47,852,342)

BCVA: best-corrected visual acuity; CES-D: Centre for Epidemiologic Studies Depression; GDS: Geriatric Depression Scale; ICD: International Classification of Disease; PHQ: Patient Health Questionnaire.

**Table 2 jcm-11-02412-t002:** Proportion of participants with depression overall and among visual impaired, and proportion of participants with visual impairment overall and among depressed.

	N. Subjects (N. Studies)	Pooled Proportion (95% CI)
Depression	929,678 (27)	0.17 (0.13–0.22)
Surveys	583,563 (17)	0.19 (0.14–0.27)
Clinical Examination	36,167 (8)	0.15 (0.10–0.23)
Administrative databases	309,948 (2)	0.07 (0.03–0.16)
*p*-value *		0.0664
Visual Impairment	48,568,108 (28)	0.10 (0.07–0.16)
Surveys	611,430 (16)	0.16 (0.10–0.25)
Clinical Examinations	64,559 (9)	0.07 (0.03–0.13)
Administrative databases	47,892,290 (3)	0.03 (0.01–0.07)
*p*-value *		0.0018
Depression among visually impaired	566,156 (21)	0.27 (0.21–0.33)
Surveys	82,804 (10)	0.29 (0.22–0.37)
Clinical Examinations	3323 (8)	0.26 (0.17–0.38)
Administrative databases	480,029 (3)	0.22 (0.08–0.46)
*p*-value *		0.7868
Visual Impairment among depressed	130,048 (20)	0.20 (0.13–0.29)
Surveys	89,897 (10)	0.28 (0.16–0.43)
Clinical Examinations	3459 (8)	0.15 (0.09–0.25)
Administrative databases	36,692 (2)	0.08 (0.02–0.33)
*p*-value *		0.1387

* *p*-value for heterogeneity between groups.

**Table 3 jcm-11-02412-t003:** Association between VI and depression.

	No. of Subjects (No. of Studies)	OR (95 % CI)	I^2^
Unadjusted OR			
Surveys	617,355 (14)	1.89 (1.51–2.37)	89%
Clinical Examination	69,178 (10)	2.17 (1.76–2.67)	66%
Administrative databases	48,162,290 (3)	3.34 (1.01–11.11)	100%
Adjusted OR			
Surveys	50,437 (8)	1.75 (1.34–2.30)	99%
Clinical Examinations	57,352 (4)	1.17 (1.00–1.37)	0%
Administrative databases	48,143,511 (2)	2.47 (0.97–6.33)	100%

## Data Availability

All data relevant to the study are included in the article or uploaded as Appendix A.

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
