# Peer review of "The Association between Vision Impairment and Depression: A Systematic Review of Population-Based Studies"

_jcm, 2022, doi:10.3390/jcm11092412_

Round 1

Reviewer 1 Report

Dear authors,

Journal of Clinical Medicine

Manuscript Number ID: jcm-1638656

Title: Association between vision impairment and depression: a systematic review of population-based studies

GENERAL CONSIDERATIONS

  • The authors have described a systematic review regarding the association between depression and visual impairment.
  • Systematic reviews are usually based on specific search strategies and methods for more reliable data (such as PRISMA or Cochrane).
  • There is also a need to answer a specific question such as “to investigate if depression has any association with visual impairment.” Please describe clearly the question of this search.

ABSTRACT

  • The authors have mentioned as purpose: “We conduct a systematic review and meta-analysis on the association between Vision Impairment (VI) and depression in population-based studies in adults.” There must have a question to be answered in the PURPOSE of this ABSTRACT. Please also be specific on the “question” on the PURPOSE. The purpose of this study seems too vague and superficial.
  • The authors have mentioned as conclusion: “Since both VI and depression are commonly associated morbidities in adults, the presence of the association between depression and VI should be considered when planning multidisciplinary intervention aimed at identifying and treating each condition. Their simultaneous occurrence could have an important impact on the well-being of affected people [64], and multidisciplinary programs targeting the treatment of both conditions should be pursued in such patients.”
  • Please do not mention sentences that cannot be supported by data shown. The first paragraph does not answer anything from the study. If IV should be considered when planning any intervention was never an issue here. Please delete this sentence and conclude something.
  • Last sentence is not part of this study. The impact of this condition was never an issue here. Please delete it and conclude something.

INTRODUCTION

  • The authors have mentioned as purpose: To our knowledge, no recent reviews or meta-analyses have been published specifically on the burden of the association between VI and depression in the general population.” Please do not claim priority unless search protocol is shown (with keywords, year and source).

MATERIAL AND METHOD

  • Study design must be mentioned in this section with detail according to Journal’s rules cited in the website.
  • Systematic reviews are usually based on specific search strategies and methods for more reliable data (such as PRISMA or Cochrane).

DISCUSSION

  • The authors have mentioned as purpose: “Our study has aimed to fill this gap as this is important to develop public health strategies to improve the quality of life of older adults.” Please be more specific on the PURPOSE of this systematic review.
  • The authors have mentioned as conclusion: “Given the relatively high prevalence and coexistence of depression and VI in the general adult population, both conditions should be actively sought out when either is diagnosed, with one in five subjects with depression also having VI on clinical examination and one in four in surveys. Their simultaneous occurrence could have an important impact on the well-being of affected people [64], and multidisciplinary programs targeting the treatment of both conditions should be pursued in such patients.”
  • Please do not mention sentences that cannot be supported by data shown. Please observe that all conclusions do not answer the purpose of the study. Please delete both sentences and conclude something based on data shown.

Author Response

Reviewer #1

GENERAL CONSIDERATIONS

The authors have described a systematic review regarding the association between depression and visual impairment.

Systematic reviews are usually based on specific search strategies and methods for more reliable data (such as PRISMA or Cochrane).

There is also a need to answer a specific question such as “to investigate if depression has any association with visual impairment.” Please describe clearly the question of this search.

ABSTRACT

The authors have mentioned as purpose: “We conduct a systematic review and meta-analysis on the association between Vision Impairment (VI) and depression in population-based studies in adults.” There must have a question to be answered in the PURPOSE of this ABSTRACT. Please also be specific on the “question” on the PURPOSE. The purpose of this study seems too vague and superficial.

Reply: We changed the sentence as follow:

“We conducted a systematic review and meta-analysis to investigate whether depression is associated with Vision Impairment (VI) in population-based studies in adults”

The authors have mentioned as conclusion: “Since both VI and depression are commonly associated morbidities in adults, the presence of the association between depression and VI should be considered when planning multidisciplinary intervention aimed at identifying and treating each condition. Their simultaneous occurrence could have an important impact on the well-being of affected people [64], and multidisciplinary programs targeting the treatment of both conditions should be pursued in such patients.”

Please do not mention sentences that cannot be supported by data shown. The first paragraph does not answer anything from the study. If IV should be considered when planning any intervention was never an issue here. Please delete this sentence and conclude something. Last sentence is not part of this study. The impact of this condition was never an issue here. Please delete it and conclude something.

Reply: In the abstract we have concluded as follows:

“In conclusion, VI and depression are prevalent morbidities and should be actively sought when either is identified, especially in older adults.”

INTRODUCTION

The authors have mentioned as purpose: “To our knowledge, no recent reviews or meta-analyses have been published specifically on the burden of the association between VI and depression in the general population.” Please do not claim priority unless search protocol is shown (with keywords, year and source).

Reply: We found no papers in PUBMED and no protocols in PROSPERO to date. We changed the sentence as follow:

“A PUBMED and PROSPERO search (((depression OR depressive) AND (vision OR visual))) revealed that, to date (30/03/2022), that there were no recent reviews or meta-analyses on the burden of the association between VI and depression in the general population.”

MATERIAL AND METHOD

Study design must be mentioned in this section with detail according to Journal’s rules cited in the website.

Systematic reviews are usually based on specific search strategies and methods for more reliable data (such as PRISMA or Cochrane).

Reply: We followed Cochrane recommendation for the conduction of our systematic review and PRISMA for its reporting. We added the following paragraph to the Materials and Methods Section:

“We performed a systematic review and meta-analysis according to the recommendations indicated by the Cochrane network [15]. The reporting was in accordance with the Preferred Reporting Items for Systematic reviews and Meta-Analyses (PRISMA) criteria [16].”

DISCUSSION

The authors have mentioned as purpose: “Our study has aimed to fill this gap as this is important to develop public health strategies to improve the quality of life of older adults.” Please be more specific on the PURPOSE of this systematic review.

Reply: In the Abstract (as Reviewer suggested in the comment 2.a.) and in the Introduction, we changed the purpose of our systematic review as follow:

“Our study aimed to investigate whether depression is associated with VI in community-dwelling adults”

The authors have mentioned as conclusion: “Given the relatively high prevalence and coexistence of depression and VI in the general adult population, both conditions should be actively sought out when either is diagnosed, with one in five subjects with depression also having VI on clinical examination and one in four in surveys. Their simultaneous occurrence could have an important impact on the well-being of affected people [64], and multidisciplinary programs targeting the treatment of both conditions should be pursued in such patients.”

Please do not mention sentences that cannot be supported by data shown. Please observe that all conclusions do not answer the purpose of the study. Please delete both sentences and conclude something based on data shown.

Reply: We have concluded as follow:

“In conclusion, one in five subjects with depression have VI on clinical assessment and one in four with survey data. These prevalent morbidities should be actively sought when either is identified, especially in older adults.”

Reviewer 2 Report

It could be acceptable that this review did not try to reveal causal relationship between depression and visual impairment. From the clinical point of view, it would be very useful to know data about:

the proportion of patients whose depression manifested after visual impairment and  the proportion of patients suffered from depression before VI occurred

could the ophthalmologists guess the depression after the development of VI and refer the patient to a psychotherapist? how many patients with severe visual impairment require psychological help?

could the management of visual impairment (eg usage of low vision aids) improve depression?

Author Response

Reviewer #2

It could be acceptable that this review did not try to reveal causal relationship between depression and visual impairment. From the clinical point of view, it would be very useful to know data about:

the proportion of patients whose depression manifested after visual impairment and  the proportion of patients suffered from depression before VI occurred

Reply: We included studies with cross-sectional design, thus, causal relationship between depression and visual impairment has not been explored. In the discussion, we added a paragraph explaining the bidirectional association between depression and visual impairment: please see reply to Academic Editor comment #3.

could the ophthalmologists guess the depression after the development of VI and refer the patient to a psychotherapist? how many patients with severe visual impairment require psychological help?

Reply: Our previous systematic review exploring the association between VI and depression in clinical settings found that 27% of VI patients attending clinic also has depression (ref 7). We have remarked that screening for depression is recommended in clinical eye settings for all patients with VI.

could the management of visual impairment (eg usage of low vision aids) improve depression?

Reply: We have added a reference to a Cochrane systematic review confirming that psychological interventions improve depression in patients with VI (ref 14), providing a justification for studies aiming to estimate the burden and characteristics of this type of multimorbidity, such as ours.
